# Render U-Net: A Unique Perspective on Render to Explore Accurate Medical Image Segmentation

**Chen Li [†], Wei Chen \*,[†] and Yusong Tan**

College of Computer, National University of Defense Technology, Changsha 410073, China;
lichen14@nudt.edu.cn (C.L.); ystan@nudt.edu.cn (Y.T.)
\* Correspondence: chenwei@nudt.edu.cn
† These authors contributed equally to this work.

**Abstract:** Organ lesions have a high mortality rate, and pose a serious threat to people's lives. Segmenting organs accurately is helpful for doctors to diagnose. There is a demand for the advanced segmentation model for medical images. However, most segmentation models directly migrated from natural image segmentation models. These models usually ignore the importance of the boundary. To solve this difficulty, in this paper, we provided a unique perspective on rendering to explore accurate medical image segmentation. We adapt a subdivision-based point-sampling method to get high-quality boundaries. In addition, we integrated the attention mechanism and nested U-Net architecture into the proposed network Render U-Net.Render U-Net was evaluated on three public datasets, including LiTS, CHAOS, and DSB. This model obtained the best performance on five medical image segmentation tasks.

**Keywords:** semantic segmentation; rendering; medical image; artificial intelligence; deep learning

## 1. Introduction

Organ diseases (such as coronavirus pneumonia and malignant cancer) are serious threats to human life. Limited by medical technology, early detection and treatment are still the most effective methods to increase the survival probability of such diseases. Recognizing the location and contours of organs through medical images is the medical image segmentation task, which is used as preprocessing. The appropriate preprocessing (such as [1,2]) is necessary to help doctors in automatic diagnosis and treatment.

When fighting COVID-19 worldwide, medical image segmentation can greatly assist doctors in the diagnosis of suspicious patients. Especially in the absence of early nucleic acid detection reagents, medical image segmentation based on artificial intelligence [3] has played an inestimable role in alleviating the spread of the epidemic. The studies of medical image segmentation have broad application prospects and great practical significance.

Early research on image segmentation focused mainly on manual segmentation or semi-automatic segmentation. The manual segmentation [4,5] relies to a large extent on the expert's empirical judgment, which is highly subjective. Therefore, manual image segmentation is so challenging that it cannot meet the requirements of modern medicine for segmentation tasks. Similarly, semi-automatic segmentation [6] still cannot lack human participation, which may involve subjectivity and uncertainty. On the contrary, the automatic image segmentation is to segment the target object from the image in a fully automatic manner without any human involvement. Naturally, automatic image segmentation [7,8] is the preference of modern medicine and analysis obtained extensively.

However, it is often difficult for medical images to be as clear as natural images. Directly using CNN-based natural image segmentation techniques to segment medical images will not obtain

good results. In 2015, Long [9] and Ronneberger [10] provided other effective ways (FCN and U-Net) to overcome this difficulty. Since then, there have been many studies [11,12] and applications [13] based on these two network architectures. Unfortunately, these models usually pay more attention to the overall characteristics of the target, but ignore the importance of boundaries in medical image segmentation. Such designs are likely to result in a high evaluation indicator (e.g., Dice and IoU), but this high indicator does not represent accurate segmentation because it does not perform well on boundary. Prediction errors at the boundaries will not have any influence on segmentation of natural images, but it is completely different when segmenting medical images. The rigor of medicine requires greater accuracy of boundaries, especially in segmentation of human organs. Even a small boundary error is very likely to cause a larger medical accident.

Therefore, there is an urgent demand for accurate segmentation for medical images. To optimize the segmentation performance at the boundary, we returned to the essence of image analysis and looked for strategies.

In the field of image analysis, images are often regarded as regular grids of pixels (points). Their feature vectors are hidden representations on the regular grid of the image. In image semantic segmentation, the pixels in the hidden representation will be mapped to a set of labels after uniform upsampling. These labels are used as output masks to indicate the predicted category at each pixel. Because the pixels in boundaries are only small percentages of the entire image, the proportion of the object interior is relatively large and smooth. Thus, the original up-sampling method will unnecessarily oversample the object interior but undersample the object boundary. Image segmentation methods often upsample regular grids on a low resolution, which is a compromise between undersampling and oversampling. This is one important reason for blurry boundary.

To solve this difficulty, we turned to the rendering method in computer graphics for help. The rendering method does not calculate the labels uniformly on the hidden representation. Rather, it adaptively selects uncertain points from all pixels of the image to calculate labels. For example, the subdivision strategy [14] can effectively render anti-aliased high-resolution images. This is the difference between rendering and upsampling.

As a consequence, we provide a unique perspective on render and propose a network based on nested UNet, named **Render U-Net**. Our work is to explore the challenges of accurate medical image segmentation, which can be summarized as follows:

1. Render U-Net draws on the idea of render in computer graphics. For "rendering" high-quality boundaries, Render U-Net adapts a subdivision-based point-sampling method to replace the original upsampling method (Section 3.3).
2. Render U-Net integrates the Attention mechanism (Section 3.2) into the nested U-Net architecture (Section 3.1), and it introduces deep supervision (Section 3.4) to supervise the extraction of semantic information at all levels. This gives trained Render U-Net a flexible structure, and so it performs pruning operations during testing (Section 3.5).
3. Render U-Net was evaluated on three public datasets (LiTS, CHAOS, and DSB), and it achieved very competitive performance in five medical image segmentation tasks (Section 4.4).
4. Although the render method has been frequently used in computer graphics, this is the first time that the main idea of rendering is introduced into the semantic segmentation for medical images.

The remainder of our paper is organized as follows: Section 2 will give an introduction to related works in this field. Section 3 will analyze the proposed methods, including nested network architecture, attention gate, point-sampling method, deep supervision, and model pruning. Section 4 will explain the experimental setups and analyzes the experimental results. Section 5 will conclude this work.

## 2. Related Works

### 2.1. U-Net Architecture and Variants

U-Net [10] is one of the most studied approaches in the field of medical image segmentation. This network has a very symmetrical encoder–decoder structure, where the encoder and decoder are on the left and right sides of the network, respectively. There are basic convolutional layer and activation layer ReLU on both sides. In addition, in the encoder, a 2 × 2 max-pooling layer is followed by the convolutional layer to increase the robustness to disturbance.In the decoder, upsampling restores the features extracted by the convolutional layer from the low-resolution feature map. This process is repeated until it is decoded to the resolution of the input. Between the encoder and decoder in U-Net, there are also skip connections to reduce the loss of semantic information caused by downsampling. Due to the above settings, U-Net became the more preferred choice when segmenting medical images.

As a consequence, the applications of U-Net are also widely common in computer vision tasks [15,16]. At the same time, many studies have improved the U-Net architecture and proposed various U-Net variants accordingly. H-DenseUNet [11] is proposed by Li et al. This variant retains effective features to the maximum extent by combining dense connections of 2D and 3D. R2U-Net [17] is proposed by Alom et al. This variant upgrades the original U-Net structure, combining the residual unit and RCNN. ResUNet-a [18] proposed by Diakogiannis is another variant based on R2U-Net. It not only adds residual connections, but also combines pyramid scene parsing pool and multi-task inference in the new network. From another perspective, Zhou et al. redesigns the skip connections between the nested convolution layers and proposes UNet++ [19]. Immediately afterwards, Li et al. added an attention mechanism to it and proposes the ANU-Net [20]. UNet3+ [21] is an improvement of connections in the original U-Net. This network combines multi-scale functions.

### 2.2. The Attention Mechanism

The attention mechanism is inspired by the study of human vision. The human visual system can collect hundreds of millions of external information every second. Therefore, in order to efficiently mine useful information, this system will automatically select the collected information and only focus on specific parts of it, while ignoring other useless parts.

Natural language processing (NLP) firstly applied the attention mechanism to the extraction of key grammatical components of sentences. The attention mechanism is used to identify key semantics (such as subject, predicate, object, etc.), and ignore descriptive semantics (such as adjective). This can help the learning process of the model more targeted. Naturally, the attention mechanism is then widely used in other fields [22–27] because it enables the model to distinguish and concentrate key information. In the semantic segmentation task, most of the research [25,28] regards the attention mechanism as a module. RA-UNet [25] is built on the U-Net, combining with residual connection and attention mechanism.

### 2.3. PointRend

PointRend [29] is a new method of image segmentation proposed by the Facebook artificial intelligence laboratory. They believe that the challenge of image segmentation can also be solved by rendering, and proposed PointRend. This module can be built on many advanced models that currently exist.

In various quantitative evaluations on the COCO and Cityscapes datasets, PointRend has achieved significant gains by improving mask AP by 1 to 2 percentage points for both instance and semantic segmentation. In addition, PointRend also depends on a backbone. The stronger the backbone, the better the segmentation performance.

## 3. Methodology

In this paper, we introduced five methodologies to solve five kinds of challenges in medical image segmentation. An overview of these methodologies is shown in Figure 1.

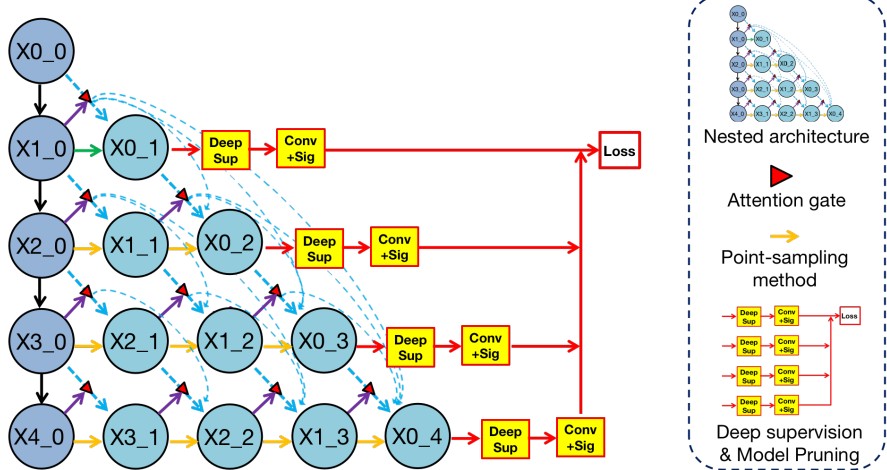

**Figure 1.** Overview of methodologies in Render U-Net. Nested architecture is analyzed in Section 3.1. Attention mechanism is analyzed in Section 3.2. The point-sampling method is analyzed in Section 3.3. Deep supervision is analyzed in Section 3.4. Model pruning is analyzed in Section 3.5.

- To collect full semantic information from different levels, Render U-Net is built on the nested U-Net architecture. This architecture can learn various features from different depths. The dense skip connections are added between convolutional blocks (Section 3.1).
- To focus on the global information of the target organ, we introduce the Attention mechanism as a module in the nested architecture (Section 3.2).
- To locate detailed boundary information of the target organ, we propose the point-sampling method to replace the original upsampling method (Section 3.3).
- To effectively supervise the semantic information extraction process at all depths, the deep supervision and new hybrid loss function are introduced. (Section 3.4).
- To reduce model parameters and apply Render U-Net to mobile devices, we remove unrelated components during inference and get a pruned, shallower version of Render U-Net (Section 3.5).

### 3.1. Nested U-Net Architecture

U-Net and its variants are the popular backbone in semantic segmentation experiments. UNet++ [19] and ANU-Net [20] are two representative architectures. The nested U-Net architecture of Render U-Net referred to these works. The structure of the Render U-Net is shown in Figure 2.

We see in Figure 2 that the structure of Render U-Net has perfect symmetry, showing an inverted pyramid shape. In this inverted pyramid structure, the convolutional blocks are nested from top to bottom. This nested structure design allows the network to train only one shared encoder for feature extraction and four decoders for feature recovery.

In the encoder, convolutional blocks transfer features from top to bottom. In the decoders, the detailed information is restored by point-sampling from the deeper layer in a bottom–up manner. This point-sampling is a kind of rendering method, which predicts the mask from low resolution to high resolution by selecting uncertain points. Furthermore, dense skip connection under attention selection is set up between each convolution block, which strengthens the effectiveness of semantic information transmission. In this way, concatenation layers fuse the semantic information from different levels.

The nested architecture can extract features of different depths. The deep supervision (Section 3.4) allows the network to spontaneously select the features that are most suitable for the task, thereby avoiding a complex selection of features. In addition, the attention mechanism (Section 3.2) and the point-sampling method (Section 3.3) can be integrated in this architecture.

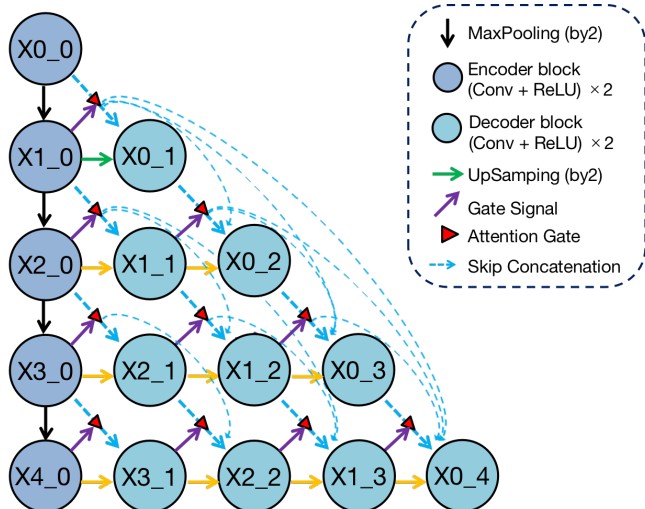

**Figure 2.** Structure of Render U-Net. Every convolutional block captures features through two convolutional layers, which are activated by ReLU functions. From a horizontal perspective, all blocks are connected by dense skip connections with attention gates. Each block in the decoders concatenates multi-scale features from its all preceding blocks. Each block in the encoder transfers features from top to bottom through the maxpooling method from a vertical perspective, and each block in the decoder integrates multi-scale features across different resolutions through the point-sampling method. In addition, the block in deeper layer is sent to the attention gate as a gate signal.

### 3.2. Attention Gate

To help the model better strengthen the learning of the target region, an attention-aware module was added to the network. This module was designed by Oktay [24] and named Attention gate. The details of the Attention gate are analyzed in Algorithm 1.

---

**Algorithm 1:** Framework of the Attention Gate in Render U-Net.

---

**Input:** g, the upsampling feature;
f, the corresponding depth feature.
**Output:** $F_3$, Attention selected feature.

1　$Result = BN_g \left( W_g^T \times g + b_g \right) + BN_f \left( W_f^T \times f + b_f \right)$.

2　Convolutional operation $(W_g, b_g)(W_f, b_f)$ extracts contextual information in the gate signal (g) and then determines the focus regions in upsampling features (f). BatchNorm ($BN_g, BN_f$) normalizes the data and reduces the jitter of the data between batches, thereby improving the training speed. These two results are added, pixel by pixel, to merge the features.

3　$F_1 = \sigma_1 (Result)$.

4　ReLU activation function ($\sigma_1(x) = max(0, x)$) is used to add nonlinear factors.

5　$F_2 = BN_\theta \left( W_\theta^T \times F_1 + b_\theta \right)$.

6　Convolutional operation $(W_\theta, b_\theta)$ extracts features in the target region. BatchNorm ($BN_\theta$) normalises the data.

7　$\alpha = \sigma_2 (F_2)$.

8　Sigmoid activation function ($\sigma_2(x) = \frac{1}{1+e^{(-x)}}$) converts the Attention coefficient $\alpha$ value into the range $[-1, 1]$.

9　$F_3 = f \times \alpha$.

10　The upsampling features (f) are multiplied, pixel by pixel, by the coefficients ($\alpha$) to change the value of certain regions, so as to achieve the purpose of paying attention to or suppressing different task regions.

11　**return** $F_3$;

---

The diagram of the Attention gate is shown in Figure 3. As seen, Attention gate is placed on the dense skip connection between convolution blocks. The upsampling feature from deeper block is used as the gate signal to focus on the target from prior block. The Attention gate performs well in selection by producing the attention coefficient $\alpha$. Multiplying $\alpha$ and the feature from the encoder pixel by pixel can change the value and distribution of the latter. To strengthen the learning of the target organ, the weight of the target area has been enhanced. Similarly, the weight of the irrelevant region is reduced, thereby suppressing the learning of the area. Therefore, Attention gate can improve the efficiency of propagating semantic information.

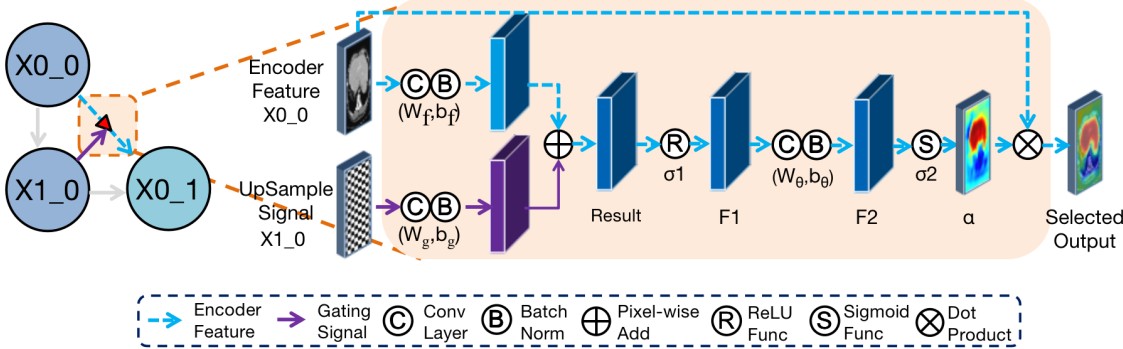

**Figure 3.** Diagram of Attention gate between convolutional block $X_{3\_0}$ and $X_{3\_1}$. The feature of $X_{4\_0}$ is used as a gate signal to focus regions in the encoder feature of $X_{3\_0}$. These two features are added after the convolution and BatchNorm layers. After activation by ReLU, the result is sent to the convolution and BatchNorm layers to extract features. Finally, the attention coefficient $\alpha$ is obtained through the sigmoid function and then it dot-products, pixel by pixel, by using the upsampling feature to get the feature after attention selection, which is $X_{3\_1}$.

### 3.3. Point-Sampling Method

Images can be regarded as regular grids of pixels (points). Their feature vectors are hidden representations on the regular grid. Pixels upsampled uniformly on the hidden representations are mapped to a set of labels. The rendering method does not calculate the labels uniformly on the hidden representations. Rather, it adaptively selects uncertain points from all pixels of the image to calculate the labels. We adopted the idea of the classic subdivision [14] strategy. This strategy calculates only those locations where the values are significantly different from their neighbors. The values at other locations are obtained by interpolating the existing results. This process, which we call the **point-sampling** method, computes masks from low resolution to high resolution by selecting uncertain points. We used the proposed point-sampling method to solve the boundary blur problem. The point-sampling method consists of the following steps:

1.  The encoder–decoder architecture takes an input image (black arrows) and extracts features from the encoder (green arrows and grids).
2.  The lightweight coarse decoder yields a coarse mask prediction for the target object (red arrows) and upsamples features (blue arrows and grids) using bilinear interpolation.
3.  To refine the coarse mask, the Point-sampling method selects some uncertain points from the coarse predicted mask, as shown by the red points in the figure.
4.  A simple MLP is set to extract the point-features of each point independently and predict their labels.
5.  The point-features are mapped to the size of the encoder feature (dashed gray arrows), and the feature at the corresponding position is replaced (dashed red arrows) to obtain the point-sampling features (red grids).
6.  The encoder-features are concatenated with the upsampling features and attention selected point-sampling features.

7.   The concatenated features are input into the fine decoder (black arrows) to obtain the fine prediction result.

From the above steps, the point-sampling method is different from the traditional upsampling method. The point-sampling method increases the selection of uncertain points (step 3) and the extraction of point-features (step 4). Therefore, the key of the point-sampling method consists of two parts. The first part is how to flexibly and adaptively select uncertain points in the image. The second part is how to extract the corresponding point-features to enhance segmentation. The detailed analysis of point-sampling method is in Figure 4.

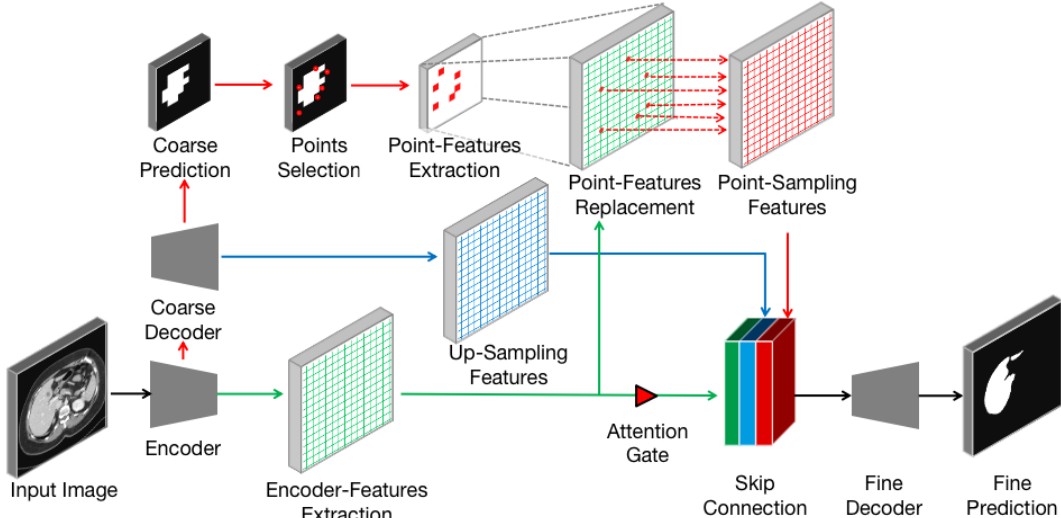

**Figure 4.** Detailed analysis of the Point-sampling method (red arrows, dots, and grids) applied to image semantic segmentation. A set of uncertain points from coarse prediction is selected to extract point-features. These features will replace original upsampling features and concatenate to refine the boundary.

**Point selection strategy.** The main idea of our selection strategy is that uncertainty points are mostly in areas where it is difficult to classify objects. After the coarse mask is predicted by coarse decoder, there are N most uncertain points on the image plane. We believe that the selection of the most uncertain points should focus not only on the uncertain areas (e.g., the boundaries between different classes), but it should also retain a certain degree of uniform coverage. Such a design can achieve the greatest degree of accurate segmentation and enhance anti-interference ability.

To achieve the above goal, we referred to the following principles in [29] for point selection and balance.

- Over generation: First, randomly sample kN points from a uniformly distributed point set. These kN points will be used to over-generate candidate points, and k should be greater than 1.
- Important sampling: Then, we will coarsely predict the segmentation class of the above kN points. After that, we will perform interpolation and calculate the uncertainty estimation of all selected points. Next, we will select the top $\beta$N points with the highest uncertainty. These points are considered important points.
- Unimportant sampling: After removing the above-mentioned important points from kN points, $(1 - \beta)$ N points remain. These points are regarded as unimportant points sampled from a uniform distribution.

In this way, we obtain the fine predicted mask from low resolution, and this procedure is shown in Figure 5.

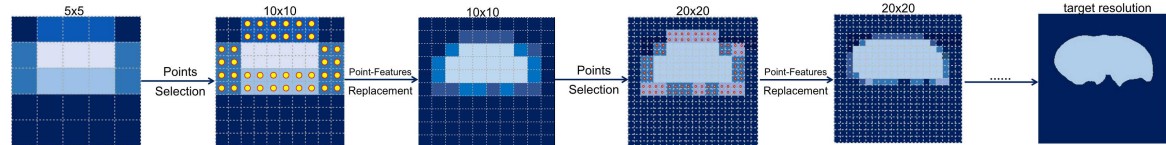

**Figure 5.** Example of the coarse-to-fine process. First, a coarse mask on a $5 \times 5$ grid is predicted by the coarse decoder. Next, the N most uncertain points (dots in grids) are selected to recover detail on the finer grid. Finally, these point-features are extracted and then used to replace the original features extracted by the encoder.

### 3.4. Deep Supervision

To improve convergence speed and solve the problem of gradient disappearance, deep supervision was first proposed in [30]. In our work, deep supervision was introduced to supervise the semantic information collected by the nested architecture. Figure 6 shows how deep supervision is implemented in Render U-Net.

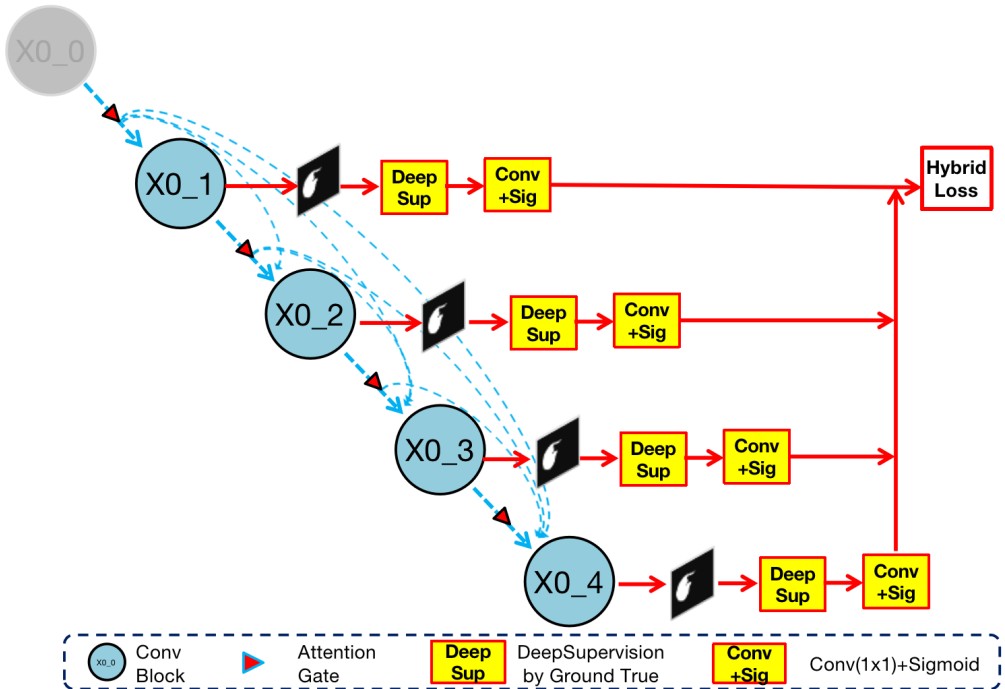

**Figure 6.** Implementation of deep supervision in Render U-Net. Ground truth supervises the output of the nested convolution structure. The four results $X_{0\_1}, X_{0\_2}, X_{0\_3}, X_{0\_4}$ are compare with ground truth and directly participate in the calculation of hybrid loss.

Because Render U-Net redesigned dense skip connections, deep supervision can help the network supervise different levels of semantic information. Then, we used the hybrid loss function in [20] to calculate the loss of the four output results from different levels. By analyzing the characteristics of medical image segmentation, the hybrid loss function reintegrated three loss functions, including dice coefficient, cross entropy, and focal loss. The mathematical formulation of the above hybrid loss function is:

$$\text{Hybrid Loss} = \sum_{i=1}^{4} \left( 1 - \left[ \frac{b \times \overline{Y} \times \log Y_i}{|Y_i - 0.5|^r} + \frac{2 \times Y_i \times \overline{Y} + s}{Y_i^2 + \overline{Y}^2 + s} \right] \right) \quad (1)$$

In the above mathematical formula, $\overline{Y}$ is the actual segmentation result after manual labeling, and $Y_i$ is the segmentation prediction result from the block $X_{0\_i}$ in the network. When calculating the loss, we regard $\overline{Y}$ and $Y_i$ as pixel matrices. All operations in the formulation are calculated on the entire

matrices. | | is the absolute value function, s is the smooth factor, $b$ is the balance factor in Focal loss, and hyper parameter $r$ is used for difficult samples.

Firstly, because of the particularity of the medical sample, positive (existing organ) samples usually occupy a small proportion in the dataset. To reduce the influence of the sample imbalance, $b$ is added as a balance factor to the above hybrid loss function, where $b$ overcomes this problem by giving different weights to different samples. This solution draws on the idea of focal loss in which the negative samples ($\overline{Y} = 1$) multiply the $b$ to get bigger loss value.

Secondly, there are also easy and difficult samples in clinical diagnosis, where simple samples mean the patients with clear symptoms while the difficult samples mean the patients with confused symptoms. In fact, there are more simple samples in the dataset. In order to better help the network to achieve a good segmentation of samples with different diagnostic difficulties, the hyper parameter $r$ is introduced. In the organ segmentation task, when predicted result $Y_i$ approaches 0, the more likely the sample is to be classified as negative. Similarly, when predicted result $Y_i$ approaches 1, the more likely the sample is to be classified as positive. On the contrary, when $Y_i$ approaches 0.5, it is so hard to diagnose that it is the difficult sample and the loss increases. As a consequence, this hyper parameter $r$ can help model optimize the segmentation of difficult samples.

Thirdly, in order to calculate the similarity between the segmentation results and the real results, the dice coefficient is introduced as a measurement, which is defined as $\frac{2 \times Y_i \times \overline{Y} + s}{Y_i^2 + \overline{Y}^2 + s}$. The higher the degree of similarity, the bigger the value of the dice coefficient.

### 3.5. Model Pruning

We have pruned the Render U-Net at four different depths and obtained four pruned sub-networks, which are shown in Figure 7. We all know that there are both input forward propagation and loss back propagation during model training. The components in the gray region can transmit errors during the loss back propagating. These components are essential for weight updates and cannot be removed. The reason why we can only remove the gray components during inference is that the network only has input forward propagation in the test phase, and the subsequent components will not affect the previous output and can therefore be removed.

As shown in the figures, all convolution blocks, upsampling, downsampling, skip connections, and attention gates in the gray area have been pruned, which are retained during training but deleted during prediction. We rename the pruned networks as Render U-Net L1, L2, L3, and L4. In addition, $LN$ also denotes that the final output of the network comes from $X_{0\_N}$. The Render U-Net selects the pruned model by evaluating the four sub-networks' segmentation performance on the validation dataset. If the performance of one sub-network on the validation dataset has reached expectations, we can use the shallower Render U-Net for segmentation during inference. For example, in the most ideal case, if the most pruned architecture L1 can have satisfactory segmentation performance when validated, then the final result of the network comes from $X_{0\_1}$. On the contrary, in the least ideal case, if only the minimally pruned architecture L4 can output good segmentation result, the final result will come from $X_{0\_4}$.

In summary, in this section, we introduced five methodologies used in Render U-Net. Section 3.1 analyzed the nested architecture to collect full semantic information from different levels. Section 3.2 analyzed the Attention gate to enhance the learning of the target organ. Section 3.3 analyzed the proposed point-sampling method to render high-quality boundaries of the target organ. Section 3.4 analyzed the deep supervision and the redesigned hybrid loss function to provide complete semantic information. Section 3.5 analyzed the model pruning to speed up prediction and decrease parameters. For the next section, we designed experiments to validate our improvements.

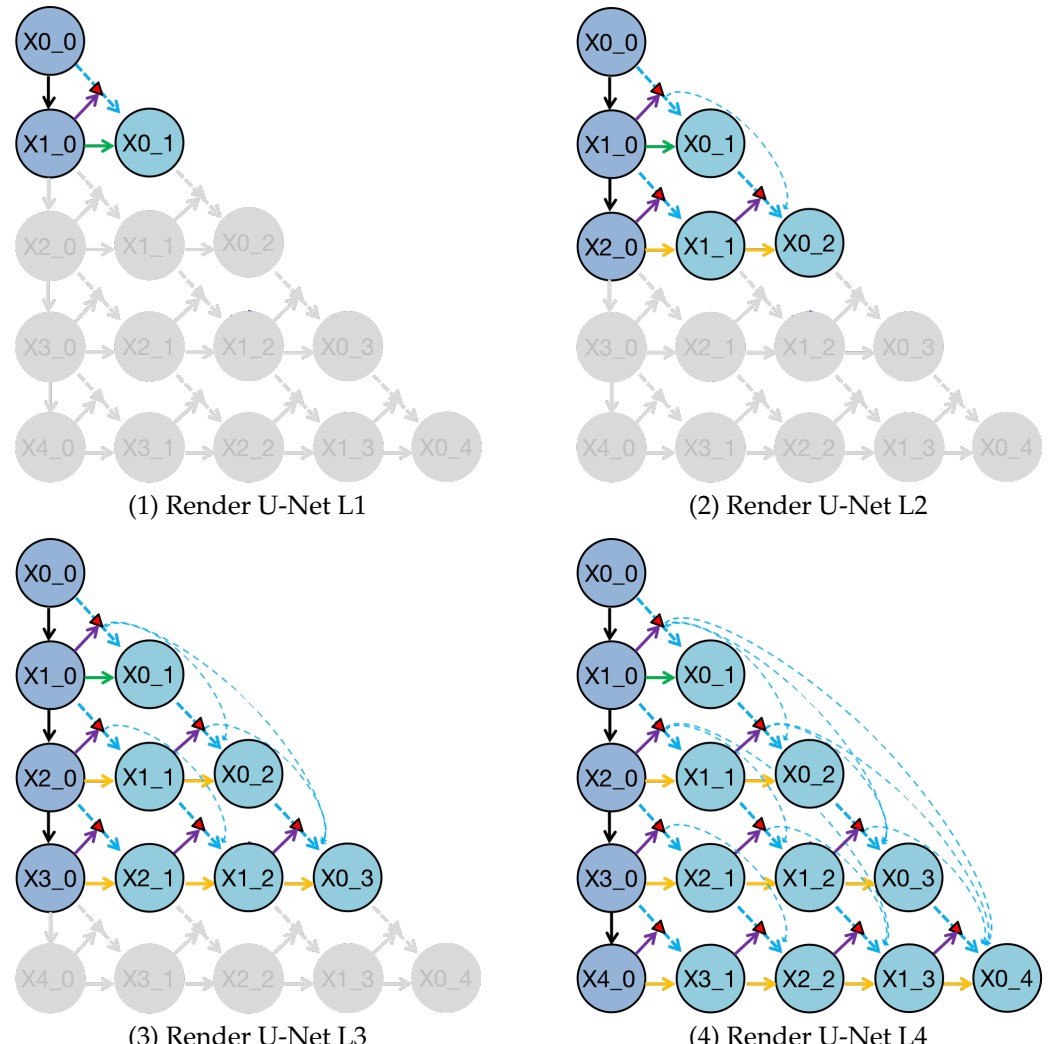

**Figure 7.** Deep supervision helps the trained Render U-Net pruning at different depths to obtain four sub-networks, which are represented by Render U-Net L1, L2, L3, and L4. The gray components in these sub-networks have been deleted during inference.

## 4. Experiments and Results

Four experiments are designed to prove our improvement due to the introduced methodologies, including segmentation experiment, attention learning experiment, point-sampling experiment, and model pruning experiment.

### 4.1. Datasets

The above four experiments are all carried out based on five public medical image datasets. The detailed information is summarized in Table 1. The target organs cover human liver, spleen, kidney and cell nuclei, and the image data come from common CT, MRI, and cell scans. The detailed introductions of the datasets are as follows:

The LiTS [31] dataset comes from the 2017 MICCAI challenge. The train set has 130 labeled CT image data and the test set has 70 unlabeled CT image data. There are three types of objects in the LiTS dataset: liver, tumor, and background. In this segmentation experiment, we only focus on the liver and regard it as a positive object, and the others as a negative object.

The CHAOS [32] dataset comes from the 2019 ISBI challenge. Only the MRI image data are used in this article, including 120 labeled DICOM datasets in the train set. There are four types of objects

in the CHAOS dataset: liver, right kidney, left kidney, and spleen. In this segmentation experiment, we considered the left and right kidney as the whole kidney and train three kinds of organs separately.

The DSB [33] dataset contains 670 labeled nuclei scans. In DSB, we first rescaled the intensity (set min at 0 and max at 255) and transformed all the images into gray images. Then, we used the Otsu threshold [34] and inverted the intensity of images so the majority of pixels were above the threshold. Moreover, we did not use any clustering, thinking that training a single model would lead to better generalization.

In order to remove irrelevant details in the image, we only take the Hounsfield Unit value in the range of $[-200, 200]$ in the image. We take one-fifth of all annotated cases as the validation dataset, and the rest as the training dataset.

**Table 1.** An overview of some important information of the medical image datasets used in this experiment.

| Application | Resolution | Modality | Provider |
|:---:|:---:|:---:|:---:|
| Liver | $256 \times 256$ | CT | 2017 MICCAI LiTS |
| Spleen | $256 \times 256$ | MRI | 2019 ISBI CHAOS |
| Kidney | $256 \times 256$ | MRI | 2019 ISBI CHAOS |
| Liver | $256 \times 256$ | MRI | 2019 ISBI CHAOS |
| Nuclei | $96 \times 96$ | Mixed | 2018 DSB Kaggle |

*4.2. Optimization Techniques and Performance Evaluation*

The proposed model is an end-to-end model, which means that we did not use too many preprocessing methods and denoising techniques. We just modified the size of the input image to meet the requirements of the network. The entire project is based on pytorch for deep learning programming and application. We used Nvidia GeForce GTX1080Ti for network training. During training, we used the hybrid loss function proposed in Section 3.4 for back propagation and network parameters update. In addition, Adam is selected as the optimizer, the learning rate is initially 0.01, and decreases to 0.0001 with training. In the optimizer, we also added a braking mechanism. When the training loss has been 20 epochs and no longer decreases, we will interrupt the training.

In the organ segmentation experiment, we used five popular indicators in medical image segmentation tasks to evaluate segmentation quality, including dice coefficient (Dice) [35], mean intersection over union (mIoU) [36] , precision [37], recall [38], and Hausdorff distance [39]. mIoU is the mean value of IoU obtained in the segmentation experiment for each class in the current dataset. The mathematical expressions of the above metrics are:

$$Dice = \frac{2 \times TP}{2 \times TP + FP + FN} \tag{2}$$

$$IoU = \frac{TP}{TP + FP + FN} \tag{3}$$

$$Precision = \frac{TP}{TP + FP} \tag{4}$$

$$Recall = \frac{TP}{TP + FN} \tag{5}$$

$$Hausdorff\ Distance(Y, \bar{Y}) = \max\{d_{Y\bar{Y}}, d_{\bar{Y}Y}\} = \max\left\{\max_{y_1 \in Y} \min_{y_2 \in \bar{Y}} d(y_1, y_2), \max_{y_2 \in \bar{Y}} \min_{y_1 \in Y} d(y_1, y_2)\right\} \tag{6}$$

where $TP$ is the True Positive, $FP$ is the False Positive, $FN$ is the False Negative. $Y$ means the segmentation result when $\bar{Y}$ is the annotated result. The $d()$ means the calculation of Manhattan distance. The Hausdorff distance is added for evaluation of the segmentation on boundary. After the mask output by the model is obtained, the pixel value of the organ area is 1, and the pixel value of the

background area is 0. When calculating the Hausdorff distance, we only consider the boundary of the object. In addition, the $\in$ denotes the boundary of the object. As the values of the first four indicators rise, the similarity increases, and the accuracy of segmentation improves.

*4.3. Feature Visualization*

In Sections 3.1 and 3.4, we introduced the nested convolution structure and dense skip connection into Render U-Net. In this section, we display the features of the first layer, which intuitively confirms the superiority of the network structure for semantic feature extraction.

As we can see in Figure 8, we used four different networks for the liver MRI image segmentation task. Then, we visualized the features of the first layer $X_{0\_N}$ ($N \in [0, 1, 2, 3, 4]$). We can find that the feature representation of U-Net is not clear enough, and the semantic information is not completely extracted. This is because the original skip connections in U-Net are relatively simple, where the encoder features are directly connected to the decoder features after upsampling. In comparison, the features in UNet++ and Render U-Net are connected by more complex skip connections, which merged more semantic information, and gradually formed a clearer feature representation. In addition, we find that, because of the introduction of deep supervision (Section 3.4), the loss of each output node could be calculated. This helps Render U-Net extract semantic information better.

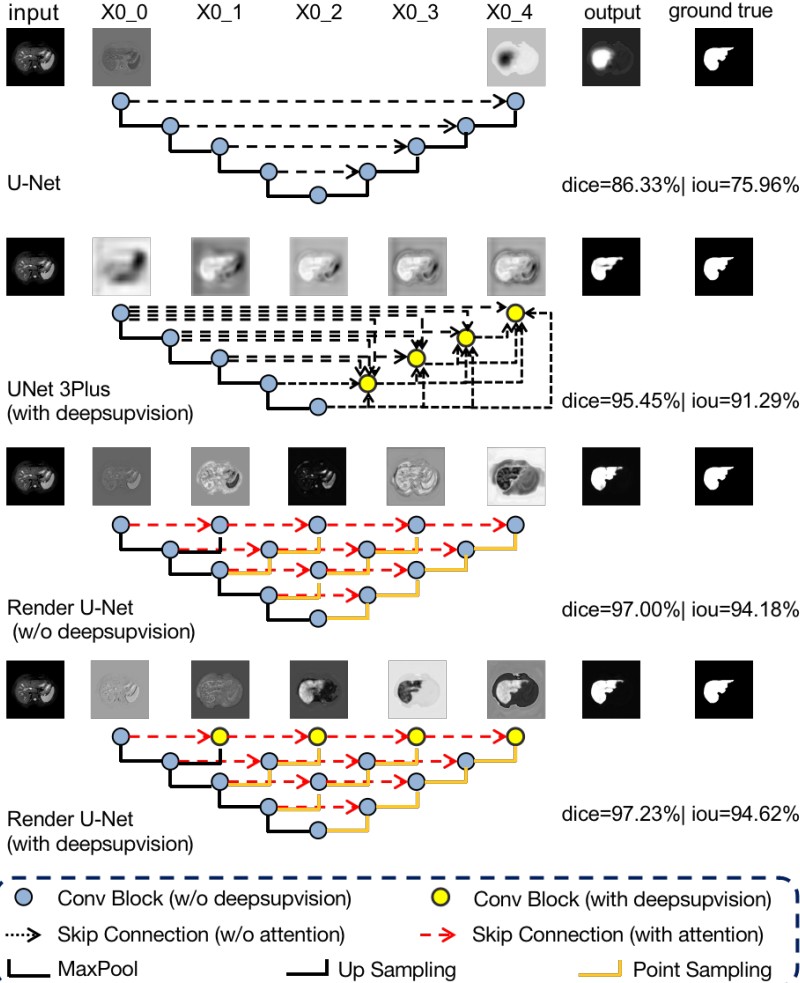

**Figure 8.** Visualization and comparison of feature maps along skip connections for liver MRI images.

*4.4. Segmentation Results*

The proposed model has completed the organ segmentation experiments under the mentioned datasets. The experimental results are compared with seven other popular models. In addition, we also combined the proposed point sampling method with UNet++ to obtain Point UNet++. This experimental result was added to the comparison to prove that the point sampling method improves the segmentation performance. Tables 2–6 collect the experimental results based on the above nine models.

**Table 2.** The experiment results of the LiTS test dataset.

| Models | mIoU | Dice | Precision (%) | Recall (%) |
|---|---|---|---|---|
| UNet [10] | 0.8949 | 0.9445 | 93.24 | 95.70 |
| R2UNet [17] | 0.9069 | 0.9511 | 93.80 | 96.48 |
| UNet++ [40] | 0.9446 | 0.9715 | 98.16 | 96.17 |
| PointUNet++ [1] | 0.9655 | 0.9825 | 98.04 | 98.45 |
| UNet3+ [21] | 0.9784 | 0.9891 | 98.94 | 98.87 |
| AttentionUNet [24] | 0.9339 | 0.9658 | 96.79 | 96.37 |
| AttentionR2UNet [2] | 0.9238 | 0.9603 | 97.12 | 94.98 |
| ANU-Net [20] | 0.9748 | 0.9815 | 98.15 | 99.31 |
| **Render U-Net** | **0.9823** | **0.9911** | **98.89** | **99.33** |

[1] PointUNet++ is the integration of UNet++ and the Point-sampling method. [2] AttentionR2UNet is the integration of R2UNet and the attention mechanism.

Table 2 compares the liver segmentation performance on CT images, where Render U-Net had the best performance. The output of model and the manually annotated result are compared in Figure 9. Our proposed network increased IoU by over 0.0484 [24], and increased the Dice coefficient by 0.0253, increased precision by 2.1 percentage points, and increased the recall rate by 2.96 percentage points.

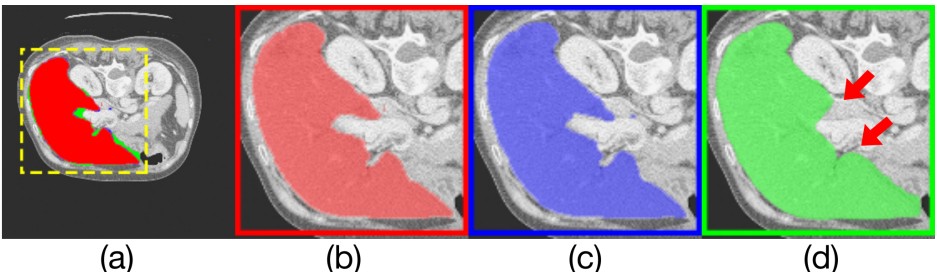

**Figure 9.** Comparison of model outputs and ground truth. (**a**) shows the other three figures stacked from top to bottom, where the predicted extra areas are clearly seen. The red area in (**b**) is the manually annotated result. The blue region in (**c**) is the output of Render U-Net. The green region in (**d**) is the output of Attention U-Net. Some areas with incorrect predictions are indicated with red arrows in the figure.

Similarly, Table 3 compares the spleen segmentation performance on MRI images, where Render U-Net had the best performance. The output of the model and the manually annotated result are compared in Figure 10. Our proposed network increased mIoU by over 0.1672 [10], and increased the Dice coefficient by 0.0981, increased precision by 14.98 percentage points, and increased the recall rate by 4.29 percentage points.

**Table 3.** Performance of spleen segmentation for MRI images.

| Models | mIoU | Dice | Precision (%) | Recall (%) |
|---|---|---|---|---|
| UNet [10] | 0.7618 | 0.8648 | 82.34 | 91.06 |
| R2UNet [17] | 0.8150 | 0.8977 | 93.60 | 86.24 |
| UNet++ [40] | 0.8105 | 0.8953 | 86.37 | 92.93 |
| PointUNet++ [1] | 0.9057 | 0.9502 | **97.70** | 92.50 |
| UNet3+ [21] | 0.8582 | 0.9235 | 94.77 | 90.18 |
| AttentionUNet [24] | 0.8413 | 0.9138 | 91.54 | 91.22 |
| AttentionR2UNet [2] | 0.8162 | 0.8979 | 90.88 | 88.73 |
| ANU-Net [20] | 0.8923 | 0.8979 | 95.19 | 93.44 |
| **Render U-Net** | **0.9290** | **0.9629** | 97.32 | **95.35** |

[1] PointUNet++ is the integration of UNet++ and the Point-sampling method. [2] AttentionR2UNet is the integration of R2UNet and the attention mechanism.

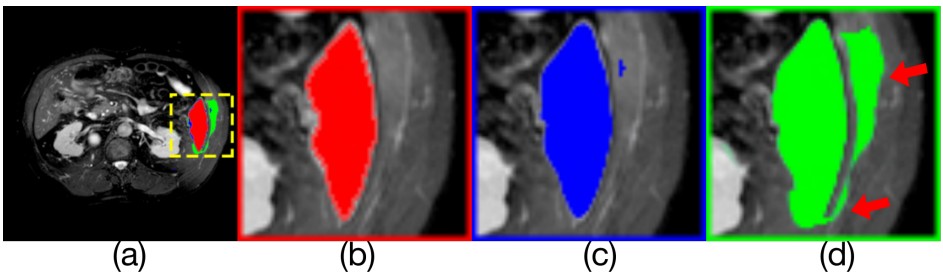

**Figure 10.** Comparison of model outputs and ground truth. (**a**) is composed of the other three figures above are stacked from top to bottom, where the predicted extra areas are clearly seen. (**b**) is the manually annotated result. The blue region in(**c**) is the output of Render U-Net. The green region in (**d**) is the output of U-Net. Some areas with incorrect predictions are indicated with the red arrows in the figure.

Table 4 compares the liver segmentation performance on MRI images, where Render U-Net had the best performance. The output of model and the manually annotated result are compared in Figure 11. Our proposed network increased IoU by over 0.0267 [20], increased the Dice coefficient by 0.0149, increased the precision by 1.3 percentage points, and increased the recall rate by 1.68 percentage points.

**Table 4.** Performance of liver segmentation for MRI images.

| Models | mIoU | Dice | Precision (%) | Recall (%) |
|---|---|---|---|---|
| UNet [10] | 0.7537 | 0.8596 | 87.31 | 84.65 |
| R2UNet [17] | 0.7780 | 0.8750 | 92.11 | 83.39 |
| UNet++ [40] | 0.8423 | 0.9139 | 93.06 | 89.79 |
| PointUNet++ [1] | 0.8551 | 0.9218 | 94.01 | 90.42 |
| UNet3+ [21] | 0.8224 | 0.9005 | 93.00 | 88.02 |
| AttentionUNet [24] | 0.7600 | 0.8637 | 91.11 | 82.09 |
| AttentionR2UNet [2] | 0.8244 | 0.9035 | 94.74 | 86.40 |
| ANU-Net [20] | 0.8789 | 0.9355 | 94.23 | 92.88 |
| **Render U-Net** | **0.9056** | **0.9504** | **95.53** | **94.56** |

[1] PointUNet++ is the integration of UNet++ and the Point-sampling method. [2] AttentionR2UNet is the integration of R2UNet and the attention mechanism.

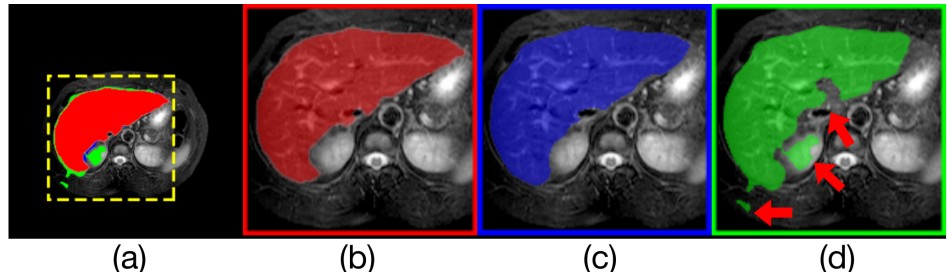

**Figure 11.** Comparison of model outputs and ground truth. (**a**) is composed of the other three figures above stacked from top to bottom, where the predicted extra areas are clearly seen. The red area in (**b**) is the manually annotated result. The blue region in (**c**) is the output of Render U-Net. The green region in (**d**) is the output of ANU-Net. Some areas with incorrect predictions are indicated with the red arrows in the figure.

Table 5 compares the kidney segmentation performance on MRI images, where Render U-Net had the best performance. The output of model and the manually annotated result are compared in Figure 12. Our proposed network increased IoU by over 0.038 [40], and increased the Dice coefficient by 0.0213, increased precision by 4.23 percentage points, and increased the recall rate by 0.95 percentage points.

**Table 5.** Performance of kidney segmentation for MRI images.

| Models | mIoU | Dice | Precision (%) | Recall (%) |
|---|---|---|---|---|
| UNet [10] | 0.8446 | 0.9158 | 89.20 | 94.08 |
| R2UNet [17] | 0.8554 | 0.9221 | 91.92 | 92.50 |
| UNet++ [40] | 0.8658 | 0.9281 | 90.87 | 94.82 |
| PointUNet++ [1] | 0.8942 | 0.9442 | 92.66 | **96.24** |
| UNet3+ [21] | 0.8449 | 0.9155 | 89.91 | 93.44 |
| AttentionUNet [24] | 0.8577 | 0.9234 | 90.97 | 93.76 |
| AttentionR2UNet [2] | 0.8734 | 0.9324 | 91.41 | 95.15 |
| ANU-Net [20] | 0.9010 | 0.9479 | 94.00 | 95.60 |
| **Render U-Net** | **0.9038** | **0.9494** | **94.14** | 95.77 |

[1] PointUNet++ is the integration of UNet++ and the Point-sampling method. [2] AttentionR2UNet is the integration of R2UNet and the attention mechanism.

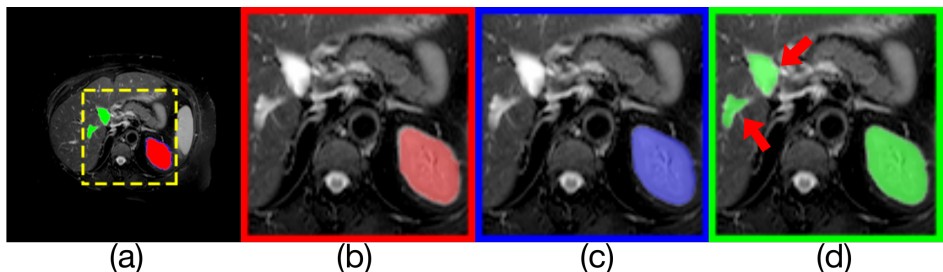

**Figure 12.** Comparison of model outputs and ground truth. (**a**) is composed of the other three figures above stacked from top to bottom, where the predicted extra areas are clearly seen. The red area in (**b**) is the manually annotated result. The blue region in (**c**) is the output of Render U-Net. The green region in (**d**) is the output of UNet++. Some areas with incorrect predictions are indicated with red arrows in the figure.

Table 6 compares the nuclei segmentation performance, where Render U-Net had the best performance. The output of model and the manually annotated result are compared in Figure 13. Our proposed network achieves IoU that increased over 0.1504 [21], and the Dice coefficient increased by 0.0951, the precision decreased 11.68 percentage points, and the recall rate increased 8.18

percentage points. It is also worth noting that U-Net also performs well in nuclei segmentation tasks. In particular, because U-Net has a simpler model, it is more suitable for such task.

**Table 6.** Performance of nuclei segmentation by the models in the DSB test dataset.

| Models | mIoU | Dice | Precision (%) | Recall (%) |
|---|---|---|---|---|
| UNet [10] | 0.7909 | 0.8833 | 86.18 | 90.59 |
| R2UNet [17] | 0.6987 | 0.8199 | 81.23 | 82.77 |
| UNet++ [40] | 0.7028 | 0.8229 | 92.36 | 74.36 |
| PointUNet++ [1] | 0.7507 | 0.8569 | 92.22 | 80.06 |
| UNet3+ [21] | 0.6853 | 0.8121 | 81.20 | 81.28 |
| AttentionUNet [24] | 0.7660 | 0.8675 | 85.81 | 87.71 |
| AttentionR2UNet [2] | 0.7025 | 0.8235 | 89.85 | 76.16 |
| ANU-Net [20] | 0.8259 | 0.9003 | 91.42 | **89.77** |
| **Render U-Net** | **0.8360** | **0.9072** | **92.88** | 89.46 |

[1] PointUNet++ is the integration of UNet++ and the Point-sampling method. [2] AttentionR2UNet is the integration of R2UNet and the attention mechanism.

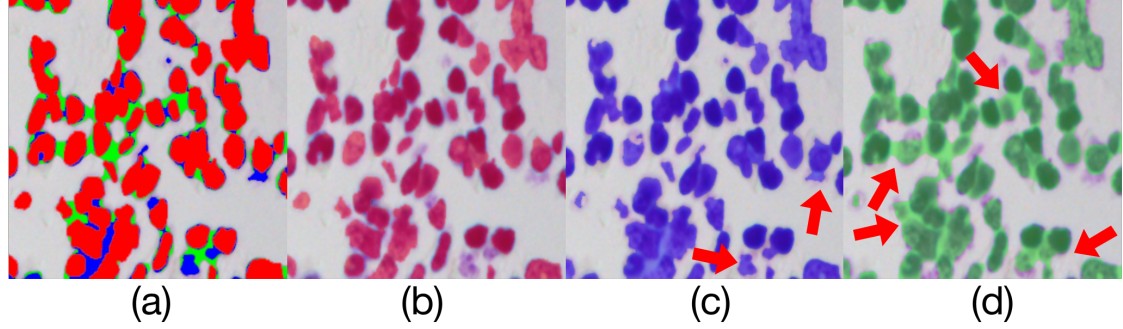

**Figure 13.** Comparison of model outputs and ground truth. (**a**) is composed of the other three figures above stacked from top to bottom, where the predicted extra areas are clearly seen. The red area in (**b**) is the manually annotated result. The blue region in (**c**) is the output of Render U-Net. The green region in (**d**) is the output of UNet3+. Some areas with incorrect predictions are indicated with red arrows in the figure.

Obviously, the proposed network outperformed other networks in the five segmentation experiments above. The improvement is attributed to the nested structure, attention mechanism, point-sampling method, and deep supervision.

*4.5. Attention Learning Results*

According to the results in Table 2 through Table 6, networks based on the attention mechanism consistently outperform original networks in the five tasks. Moreover, the proposed Render U-Net outperforms any other attention-aware models. Thus, we can conclude that the introduction of the attention gate is attributed to the improvement on the segmentation performance.

Figure 14 is two groups of changes of the attention coefficient $\alpha$ when segmenting liver from the CT image on the LiTS dataset. We can get the following conclusions of the attention coefficient $\alpha$ from the figure:

- The attention coefficient will gradually change with model training. At the same time, the attention coefficient changes the weights of the focused area during training. In addition, then the attention coefficient is multiplying the feature pixel by pixel.
- The weight around the target organ will gradually increase to enhance the learning of this area, as shown in the red area in the above figure.
- Similarly, the weight of the confused tissue area that is not related to the segmentation target will gradually decrease, which is used to inhibit the learning of this area, as shown in the blue area in the figure above.

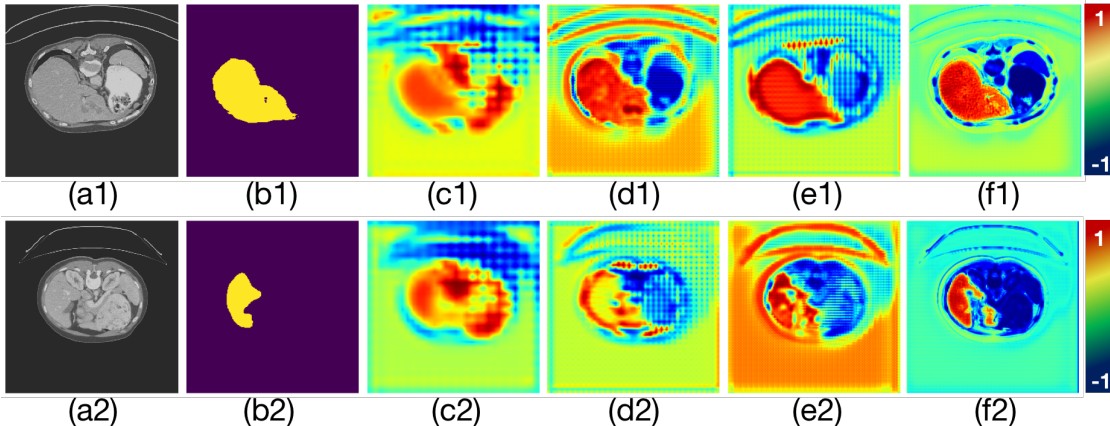

**Figure 14.** The graph of the attention coefficient $\alpha$ changing with training. The color change in the figures represents the change in the weight of attention learning, where red represents enhancement and blue represents inhibition. (**a1–f1**) is the first group of changes. (**a2–f2**) is the second group of changes. (**a1,a2**) are the input CT images. (**b1,b2**) are the ground truth of liver, respectively. (**c1–f1**) and Figure (**c2–f2**) are two groups of the attention coefficients $\alpha$ in different training phases.

*4.6. Point-Sampling Results*

We used the parameters ($k = 3$, $\beta = 0.75$, $n = 8096$) for uncertain points selection strategy. Considering that all the segmentation tasks in the experiments were binary classification tasks, we used the distance between the probability of coarse prediction mask and 0.5 as a measure of point uncertainty. Table 7 compares the boundary segmentation performance measured by Hausdorff distance, where Render U-Net had the best performance. Figure 15 shows the example of the point-sampling process for nuclei segmentation.

**Table 7.** Hausdorff distance of five segmentation tasks.

| Models | Liver (CT) | Spleen | Kidney | Liver (MRI) | Nuclei |
|---|---|---|---|---|---|
| UNet [10] | 14.95 | 11.94 | 11.87 | 30.98 | 35.54 |
| R2UNet [17] | 17.76 | 9.05 | 12.71 | 28.05 | 41.19 |
| UNet++ [40] | 9.31 | 7.82 | 11.97 | 25.59 | 34.05 |
| PointUNet++ [1] | 7.62 | 6.32 | 10.53 | 23.12 | 32.32 |
| UNet3+ [21] | 6.36 | 7.25 | 13.13 | 23.97 | 37.21 |
| AttentionUNet [24] | 7.21 | 6.23 | 13.63 | 23.09 | 34.85 |
| AttentionR2UNet [2] | 12.26 | 8.20 | 13.87 | 20.33 | 35.51 |
| ANU-Net [20] | 5.48 | 5.97 | 10.64 | 18.06 | 32.04 |
| **Render U-Net** | **4.41** | **5.39** | **10.49** | **16.96** | **31.65** |

[1] PointUNet++ is the integration of UNet++ and the Point-sampling method. [2] AttentionR2UNet is the integration of R2UNet and the attention mechanism.

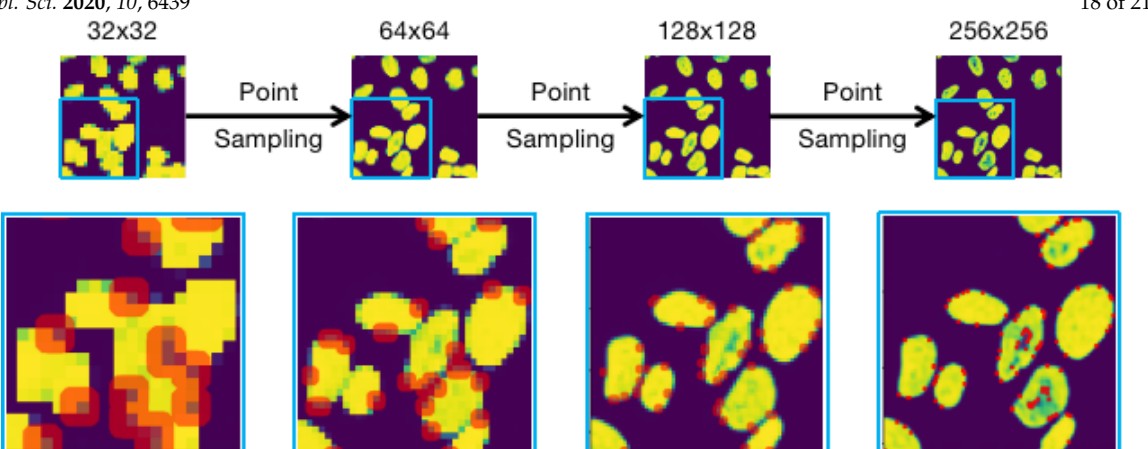

**Figure 15.** Diagram of the Point-sampling process for rendering a higher resolution boundary when inferring nuclei. The red dots in different resolution are the selected uncertain points to calculate.

As we can see, during point-sampling, our network selected a set of uncertain points and calculated masks from low-resolution ($32 \times 32$) to high-resolution ($256 \times 256$). Compared with the traditional upsampling method that calculates full-resolution points ($256 \times 256 = 64$ K), the point-sampling method can extract boundary information by using only a few points for prediction ($8096 \times 3 = 24$ K), thereby assisting the overall segmentation task and reducing computational overhead.

### 4.7. Model Pruning Results

Table 8 quantifies the segmentation performance and inference time on four tasks, including spleen segmentation, kidney segmentation, liver segmentation, and nuclei segmentation. We can get conclusions that model pruning can be used to obtain a model with fewer model parameters, which should have a faster prediction speed. More specifically, in the spleen segmentation task, the performances of Render U-Net L3 and L4 are very close, but the parameter amount of L3 is only 2.91 M. Therefore, in this task, we can use a shallow sub-network Render U-Net L3 to get segmentation performance similar to L4, which owns 12.0 M parameters. Through this pruning operation, we can save 75.75% of the parameter, and reduce the prediction time by 22.93%, while Dice only reduces 0.0178 and mIoU reduces 0.0306. In the kidney MRI image task and liver MRI image segmentation task, pruned models achieved the similar improvement on prediction speed at the expense of a small amount of performance.

**Table 8.** Segmentation results of four pruned models. The inference time (counted by seconds) is calculated by segmenting 1K images.

| Model | Spleen | | | Kidney | | | Liver | | | Nuclei | | |
|---|---|---|---|---|---|---|---|---|---|---|---|---|
| | Dice | mIoU | Time | Dice | mIoU | Time | Dice | mIoU | Time | Dice | mIoU | Time |
| L1 | 0.8258 | 0.7203 | 28.54 | 0.7588 | 0.6198 | 28.37 | 0.6624 | 0.5332 | 18.10 | 0.8490 | 0.7529 | 8.74 |
| L2 | 0.9272 | 0.8655 | 44.95 | 0.8519 | 0.7450 | 41.77 | 0.7546 | 0.6480 | 34.63 | 0.8516 | 0.7603 | 22.74 |
| L3 | 0.9451 | 0.8984 | 59.88 | 0.9353 | 0.8793 | 59.09 | 0.8797 | 0.8170 | 50.28 | 0.9072 | 0.8360 | 40.35 |
| L4 | 0.9629 | 0.9290 | 77.69 | 0.9494 | 0.9038 | 80.32 | 0.9056 | 0.9504 | 64.79 | 0.8864 | 0.8065 | 61.67 |

In particular, it is also worth noting that Render U-Net L3 performed better than L4 in the nuclei segmentation task. After analysis, we believe that this is because the semantic information extracted by L3 is enough to restore the characteristics of the nuclei, while the features extracted by L4 are more complex and not suitable for such a task. This result is mutually confirmed with the results in Table 6.

Figure 16 directly compares the relationship between performance and inference time of four pruned models with different parameters. As we can see, the parameters of the four sub-networks are various, where the parameters of the shallowest sub-network is 0.1 M, while the deepest one has 12.0 M.

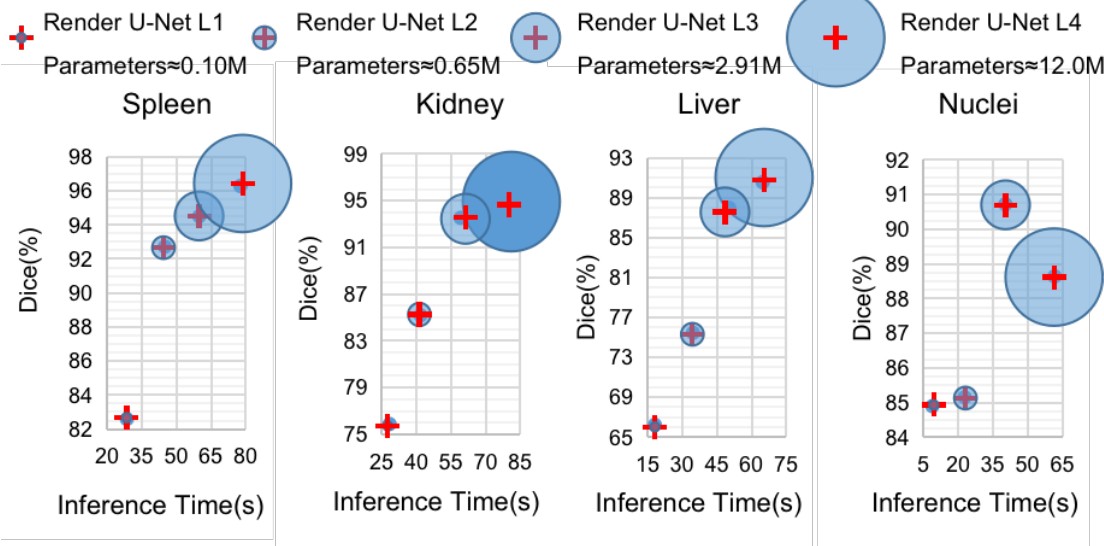

**Figure 16.** Diagram of the relationships between the Dice coefficients and inference times of the four pruned Render U-Nets with segmentation of four kinds of objects. We use four blue circles with different areas to denote sub-networks with different parameter amounts.

In the most ideal case, if the segmentation result of Render U-Net L1 has reached expectations, then we can save 99.9% of the parameters. Unfortunately, according to the four segmentation experimental results on CHAOS and DSB, the segmentation performance of L1 cannot meet the basic requirement. We analyze the reason because it is too shallow for extracting effective features.

## 5. Conclusions

In this paper, we provided a unique perspective on render, and we viewed the medical image segmentation task as a render problem. We adapted a subdivision-based point-sampling method to replace the original upsampling method for "rendering" high-quality boundaries. The proposed model, called Render U-Net in this paper, performed segmentation experiments on LiTS, CHAOS, and DSB datasets together with other models, including U-Net, R2U-Net, UNet++, Point UNet++, UNet3+, Attention U-Net, Attention R2U-Net, and ANU-Net. The experiment's results demonstrated that the proposed network outperforms in overall and boundary segmentation. We make the following analysis: the improvement in accuracy is due to the integration of the point sampling method, supervised dense skip connection, and attention mechanism in the nested architecture. The point-sampling method has been proven to render higher-quality boundaries than the traditional upsampling method.

**Author Contributions:** C.L.: Methodology and Writing—original draft; W.C.: Supervision and Writing—review and editing; Y.T.: Project administration. All authors have read and agreed to the published version of the manuscript.

**Funding:** This research was funded by the National Key Research and Development Program of China (No. 2018YFB0204301).

**Conflicts of Interest:** The authors declare no conflict of interest.

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
