# Peer review of "Render U-Net: A Unique Perspective on Render to Explore Accurate Medical Image Segmentation"

_applsci, doi:10.3390/app10186439_

Round 1
Reviewer 1 Report
The paper concerns artificial intelligence in application to preprocessing (segmentation) of medical images. The topic is important and the interest in this topic is high both in scientific community and in medical engineering. It is neccessary, however, to introduce minor corrections, additions and clarifications. First of all, the authors specified only as maoivation: "the medical image segmentation task can assist physicians in treatment as pre-diagnosis." That is true, but it should be also mentioned that the appropriate preprocesing in necessary for structural methods of image analysis that are used for automatic diagnosis - see, for instance:
Bielecka M., "Syntactic-geometric-fuzzy hierarchical classifier of contours with application to analysis of bone contours in X-ray images", Applied Soft Computing, vol.69 (2018), 368-380.
Bielecka et al. "The shape language in application to the diagnosis of cervical vertebrae pathology", PloS One, vol.13(10), article number: e0204546.
The following should be precised:
1. How Ybar and Yi in formula (1) are calculated.
2. Lines 303 and 304: Are the segmantetion result and the annotated result pixel matrices (or other sets of pixels)?
3. Manhattan and Hausdorff with a capital letter.
Author Response
To start with, we would like to sincerely THANK the reviewers for their time, efforts and valuable comments on the manuscript. We have carefully revised the manuscript in accordance with three reviewers’ comments.
Please see the attachment.

Reviewer 2 Report
To my consideration, the paper is worth publishing in Applied Sciences.
However, before that, the authors should clarify the following:
Is the over segmentation considered in the current approach? Please explain how over segmentation is avoided.
What is you training and validation set of the data sets you uses?
You defined IoU. Why did you used mIoU? Is mIoU the mean value across all the classes in the dataset?
Was some kind of preprocessing and denoising applied? I suggest to clearly mention this aspect in the Introduction.
Please specified if pruning was for speed or for a small model? Also, can you specify the accuracy values achieved in segmentation process? You mention “improvement in accuracy” but you need to provide some data. Can you quantify this?
Conclusion section: please clarify “…SOTA models”!
Author Response

(The authors gave the same response as above.)

Reviewer 3 Report
This work studies about the U-net based legion segmentation model for medical images and renders high-quality boundaries. The work has some meaningful results for readers. However, in my opinion, the work needs more improvement in results and evaluation parameter analysis. Furthermore, some parts are not well understood from the document and particularly on the implementation part. The detailed concerns are addressed as follows.
Cons:
- Abstract: I would suggest the author make the abstract a little short, concise, and meaningful sentences for better readability.
- The novelty of the paper is minimal. A precise description of the paper novelty in the last paragraph of “INTRODUCTION section” - will helpful.
- The presentation of paper can be significantly improved if the author can review the paper with an experienced English writer and polish it to make it more readable: such as long sentences, tense, misused punctuations, etc. The author must follow the Journal standard for paper drafting.
- It is nice that the author can exclusively explain their contribution, problem formulation, and describe the project objectives. Besides that, the paper needs more result verification and more analysis.
- The author should make sure about the consistent notation of variables throughout the paper.
- The reviewer would like to know a more detailed explanation about the optimization techniques followed in data learning. (Elaborate).
- The reviewer is not quite clear about the description provided in Fig. 14, 15, 16 and 12 (Please Elaborate). Make the figures a little bigger.
- The reviewer is not clear about the Algorithm -I. Please follow the Journal standard or follow any standard journal while writing the Algorithm (in the paper).
The presentation could also be significantly improved if the authors can do proofreading and address the above concerns on the revised paper.
Possible References to Author :
- "Low Dose Abdominal CT Image Reconstruction: An Unsupervised Learning Based Approach," 2019 IEEE International Conference on Image Processing (ICIP), Taipei, Taiwan, 2019, pp. 1351-1355.
- "Cognitive Analysis of Working Memory Load from Eeg, by a Deep Recurrent Neural Network," 2018 IEEE International Conference on Acoustics, Speech and Signal Processing (ICASSP), Calgary, AB, 2018, pp. 2576-2580.
Author Response

(The authors gave the same response as above.)

Round 2
Reviewer 2 Report
The authors have responded to all raised issues and have improved the manuscript accordingly.
Reviewer 3 Report
The author carefully has addressed many of the concerns including the proofreading at different pages.
As suggested in the 1st review, the author:
- Provided the implementation details of the model.
- Provided details about the figures/Tables/Plots for result verification.
- The presentation of the paper improved a lot after revision and looks reasonable.
This manuscript is a resubmission of an earlier submission. The following is a list of the peer review reports and author responses from that submission.